# Assessing the geographic specificity of pH prediction by classification and regression trees

Jacob Egelberg[1]*, Nina Pena[2], Rachel Rivera[2], Christina Andruk[3]

1 Department of Biochemistry, Northeastern University, Boston, Massachusetts, United States of America,
2 Department of Science, New Rochelle High School, New Rochelle, New York, United States of America,
3 Department of Biology, Iona College, New Rochelle, New York, United States of America

* jake1egelberg@gmail.com

**Data Availability Statement:** All relevant data are within the manuscript and its Supporting Information files.

**Funding:** The authors received no specific funding for this work.

## Abstract

Soil pH effects a wide range of critical biogeochemical processes that dictate plant growth and diversity. Previous literature has established the capacity of classification and regression trees (CARTs) to predict soil pH, but limitations of CARTs in this context have not been fully explored. The current study collected soil pH, climatic, and topographic data from 100 locations across New York's Temperate Deciduous Forests (in the United States of America) to investigate the extrapolative capacity of a previously developed CART model as compared to novel CART and random forest (RF) models. Results showed that the previously developed CART underperformed in terms of predictive accuracy (RRMSE = 14.52%) when compared to a novel tree (RRMSE = 9.33%), and that a novel random forest outperformed both models (RRMSE = 8.88%), though its predictions did not differ significantly from the novel tree (p = 0.26). The most important predictors for model construction were climatic factors. These findings confirm existing reports that CART models are constrained by the spatial autocorrelation of geographic data and encourage the restricted application of relevant machine learning models to regions from which training data was collected. They also contradict previous literature implying that random forests should meaningfully boost the predictive accuracy of CARTs in the context of soil pH.

## Introduction

### Soil pH mediates provisional and regulatory service availability

Soil pH, the concentration of hydrogen ions in a sample of soil, affects many critical biogeochemical processes. These biogeochemical processes drive plant growth and diversity, which are supporting ecosystem services that sustain provisioning services (food, water, lumber, and fuel output) and regulating services (air quality, climate, erosion, and water purification) [1, 2].

Soil pH affects the metabolic quotient qCO2, which measures organic substrate uptake by soil microbes [3]. In low pH soils, higher metabolic quotients indicate increased substrate utilization by microbes (due to more costly maintenance of internal pH) and reduced carbon

**Competing interests:** The authors have declared that no competing interests exist.

availability for plant growth [4, 5]. Conversely, more neutral pH soils experience less substrate depletion and are characterized by higher plant biomass [4]. Plant biomass is further impacted by the pH-dependent leaching of dissolved organic carbon (DOC) and dissolved organic nitrogen (DON) [6, 7]. DOC and DON influence growth elements such as soil nutrient retention and turnover, soil structure, moisture retention and availability, degradation of pollutants, carbon sequestration, and soil resilience [8]. Soil pH also affects the growth of microbial and fungal communities that regenerate plant-limiting soil nutrients, as well as the efficiency of extracellular microbial enzymes [9–14].

## Factors affecting soil pH

Soil pH is moderated by an array of ecological factors that are defined by local climates and topographies. Climatic factors include annual temperature range, warmest quarter precipitation, average annual precipitation, and average annual temperature [15, 16]. Topographic factors include elevation, slope, topographic wetness index, valley depth, channel network, ruggedness, aspect, moisture, silt content, carbon contents, plan curvature, profile curvature, stream power index, length-slope factor, volumetric water content, and parent materials [15–20].

The relevance of these factors to soil pH varies by geographic region. In southwestern China, differences in elevation explain at most 0.02% ($R^2 = 0.0002$) of the variance in soil pH [15], whereas they explain up to 25% ($R^2 = 0.25$) on the Tibetan Plateau [17] and 9% ($R^2 = 0.09$) in the Tianton National Forest [19]. Similarly, slope in southwestern China can only explain 2% ($R^2 = 0.02$) of fluctuations in soil pH, whereas in northeast Colorado it accounts for approximately 50% ($R^2 = \sim 0.5$) of fluctuations in pH [15, 20]. Similar disparities are observable for every climatic and topographic factor.

## pH prediction by machine learning

Because of the major role soil pH plays in shaping plant growth and diversity, an awareness of soil pH is crucial to determine the potential benefits of ecosystem provisioning and regulating services. In this respect, a model capable of accurately predicting soil pH from easily accessible data would provide a much-needed scientific tool for later studies. This model could be utilized for improving current understandings of environmental homeostasis and the identification of ecosystems rich with resources required for agricultural production.

Due to the multivariable and nonlinear factors influencing soil pH, recent research has turned to machine learning techniques for developing pH prediction models [21–24]. Specifically, classification and regression trees (CARTs) have been applied to ecological datasets [15, 22, 25–27]. CARTs are decision trees that utilize a series of (typically binary) data splits to predict categories or values [28].

But traditional limitations of CARTs, such as overfitting, have not been explored in the context of pH prediction. Neither have random forests (RFs), which are known to improve upon the predictive capacity of CART models [21]. RFs are comprised of many discrete CARTs and better predictions by averaging those of each individual regression tree. Further, RFs counter overfitting by merging traditional bootstrapping with elements of randomization during tree construction [22, 24, 29].

## Aim of the current study

We hypothesized that (a) Zhang et al.'s CART model would exhibit geographic specificity as a result of overfitting and (b) pH prediction by a novel RF would be more accurate than pH prediction by a novel CART. Regarding (b), we expected the accuracy of a novel CART's pH

predictions to approximate that of Zhang et al.'s CART model in their region of study and exhibit a similar % RRMSE of 6.9.

Our study objectives were to (a) test for geographic specificity by applying a CART model that was developed by Zhang et al. [15] in southwestern China's humid subtropical hilly regions to data from New York's Temperate Deciduous Forests, (b) compare the usefulness of CART and RF models at predicting soil pH, and (c) better understand the climatic and topographic factors that influence soil pH at different sites.

## Materials and methods

### Study area

The study area was located in New York State in the United States of America and spanned approximately 65,000 km$^2$ across four forested New York State subregions: Hudson Valley, Saratoga, Central Leatherstocking, and Finger Lakes (Fig 1). We randomly selected 25 state parks within the study area. Soil samples were collected at four random locations within each park resulting in 100 total pH measurements. Locations for soil pH testing were determined by partitioning each park into 10 equally sized subsections and selecting four at random [30]. Soil pH across the study area ranged between five and seven (S2 File).

### Soil pH testing

For pH measurement, approximately 100g of surface soil was collected from each sampling location and diluted with 130mL of deionized H2O. The mixture was vigorously inverted and rotated before being let to sit for 10 minutes until solutes sufficiently dissolved. pH was assessed with four-squared plastic pH test strips [32]. This protocol was adapted from [33].

Plastic pH test strips were an appropriate tool for pH measurement in the current context. Previous literature reports that, in moderately acidic solutions, four-squared pH test strips

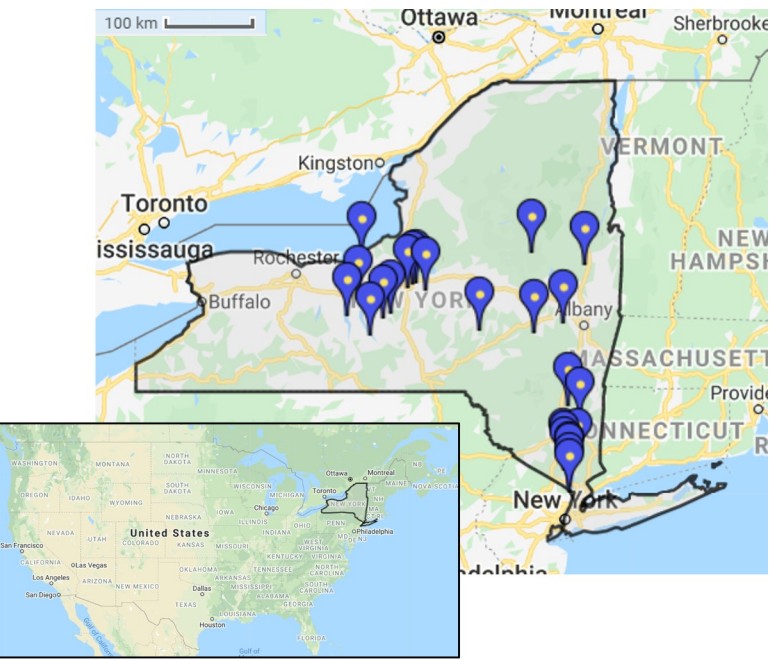

**Fig 1. Study area.** Points indicate state parks where samples were collected. Republished from [31] under a CC BY license, with permission from ZeeMaps, original copyright 2005.

exhibit positive predictive and negative predictive values greater than 95% and exceed 90% sensitivity and specificity [34]. Soil pH values observed in this study fall within this moderately acidic range, reaching five at the lowest and seven at the highest (S2 File).

## Topographic and climatic data

Topographic and climatic data for pH testing locations was gathered from gridded 90m × 90m Digital Elevation Models (DEM) and the WorldClim database, respectively, using the System for Automated Geoscientific Analysis (SAGA) version 7.6.2 [35–38]. Computational restrictions required that model grids be clipped to +0.001˚ and -0.001˚ of their original size in the longitudinal and latitudinal directions prior to basic terrain analysis of sampling locations. Topographic factors analyzed include elevation, slope, topographic wetness index (TWI), valley depth, channel network, and terrain ruggedness index (TRI). Climatic parameters analyzed include annual temperature range (ATR), precipitation of the warmest quarter (PWQ), mean annual precipitation (MAP), and mean annual temperature (MAT). Factors were selected according to those in Zhang et al. [15].

## Model parameters

Regression trees were generated with the R package 'rpart' [39, 40]. To counter overfitting, trees were pruned with 10-fold cross validation (xval) and data partitioning was ceased for nodes containing fewer than 20 observations (minsplit). Random forests were generated with the R package 'randomForest' [39, 41]. In accordance with existing literature [42], the following parameters were set for model optimization: 1000 regression trees comprised the random forest (ntree); a minimum of 1 observation (nodesize) in each node was required for data splits; a maximum of 36 nodes (maxnodes) was permitted to constitute the forest ($maxnodes = \frac{Quantity\ of\ Observations}{3}$); the quantity of predictors randomly sampled from during tree construction (mtry) was tuned to minimize the model MSE; and, the seed was generated randomly (seed = 27137). R Code is available on GitHub [43].

## Statistical analysis

Spearman correlation tests were performed to determine the strength of monotonic associations between pH and each predictor [44]. P values produced by Spearman correlations were Bonferroni-corrected to adjust for multiple hypothesis testing [45, 46] and set the global type I error rate at 0.05 (α = 0.05). A correlation was considered strong if |ρ|>0.7, moderate if 0.5<|ρ|<0.7, weak if 0.3<|ρ|<0.5, and nonexistent or very weak if 0<|ρ|<0.3 [47]. Two-tailed Wilcoxon Rank Sum tests were employed to quantify statistically significant differences (α = 0.05) between predicted pH values and measured pH values [48, 49]. Relative Root Mean Square Error (RRMSE) values were used to compare CART and RF model accuracies. RRMSE values were calculated as the square root of the average squared difference between actual pH values (y) and predicted pH values (ŷ), divided by the average of the actual pH values (ȳ) [1]. RRMSE was interpreted according to previous literature [50]. Specifically, prediction accuracy was considered "excellent" when RRMSE<10%, "good" when 10%<RRMSE<20%, fair when 20%<RRMSE<30%, and poor when 30%<RRMSE.

$$RRMSE = \frac{\sqrt{\frac{\sum_{i=1}^{n}(y_i - \hat{y}_l)^2}{n}}}{\bar{y}} \qquad [1]$$

Topographic and climatic variable importance to pH prediction was measured by CART

Variable Importance (CVI) for CARTs and % IncMSE for RFs. CVI was assessed for a predictor by summing the goodness of split (GOS) measures for each split for which it was the primary predictor (PP) and goodness of split multiplied by adjusted agreement (AA) for each split for which it was a surrogate predictor (SP) [2] [51]. Percent IncMSE was calculated for each predictor by subtracting $MSE_0$ from $MSE_j$, dividing by $MSE_0$, and multiplying by 100 [3], where $MSE_0$ is the MSE of the RF model and $MSE_j$ is the MSE of the RF model after the random permuting of predictor values [52].

$$CART\ Variable\ Importance = \Sigma GOS_{PP} + \Sigma GOS_{SP} \times AA \qquad [2]$$

$$\%IncMSE = \frac{MSE_j - MSE_0}{MSE_0} \times 100 \qquad [3]$$

Larger relative CVI and % IncMSE values indicate greater variable importance to pH prediction.

Model generation and statistical analysis were conducted in R version 4.0.0 [39]. Predictions by Zhang et al.'s CART model were calculated manually in Microsoft Excel.

## Results

### Descriptive statistics

Sampling location elevation ranged from 2m above sea level to 547m and averaged 190.5m, with land sloping between 0° and 44.7°. Precipitation and temperature measurements were similarly variable; precipitation measured between 870mm and 1292mm and temperature between 4.4°C and 11.36°C. Additional statistics are available in Table 1 and S2 File.

Spearman correlation tests provided the strength of the monotonic association between climatic and topographic variables and soil pH (Table 2). Bonferroni-adjusted p values were referenced to determine significance. There were moderate, significant, and negative associations between Mean Annual Precipitation (MAP) and soil pH ($\rho$ = -0.51; p<0.001), and Precipitation of the Warmest Quarter (PWQ) and soil pH ($\rho$ = -0.51; p<0.001). Weak significant associations were observed between soil pH and Slope ($\rho$ = -0.31; p = 0.017) and soil pH and Terrain Ruggedness Index (TRI) ($\rho$ = -0.33; p = 0.008). All other associations were nonexistent or very weak.

**Table 1. Descriptive statistics for topographic and climatic factors across study area.**

|  | Factors | Minimum | Maximum | Mean | Standard Deviation |
|---|---|---|---|---|---|
| *Topographic* | Elevation | 2 | 547 | 190.5 | 140.2 |
|  | Slope | 0 | 44.7 | 5.8 | 5.9 |
|  | TWI | 4.46 | 11.34 | 7.6 | 1.1 |
|  | Valley depth | 0 | 52 | 7.6 | 9.6 |
|  | Channel network | 0 | 546.48 | 182.1 | 140.5 |
|  | TRI | 0 | 46.77 | 6.7 | 6.1 |
| *Climatic* | ATR | 35.1 | 41 | 36.8 | 1.2 |
|  | PWQ | 256 | 335 | 295.7 | 19.7 |
|  | MAP | 870 | 1292 | 1090.6 | 111.7 |
|  | MAT | 4.4 | 11.36 | 8.8 | 1.7 |

Abbreviations defined as follows: topographic wetness index (TWI), terrain ruggedness index (TRI), annual temperature range (ATR), precipitation of the warmest quarter (PWQ), mean annual precipitation (MAP), and mean annual temperature (MAT). Reference S1 File for a definition of each factor.

**Table 2. Spearman correlations between factors and soil pH.**

| | Factor | ρ | p value | Adjusted p value |
|---|---|---|---|---|
| *Topographic* | Elevation | 0.2110 | 0.0351 | 0.3511 |
| | Slope | -0.3096 | 0.0017 | 0.0172* |
| | TWI | 0.2943 | 0.0029 | 0.0295* |
| | Valley depth | -0.1644 | 0.1020 | 1.0000 |
| | Channel network | 0.2224 | 0.0262 | 0.2617 |
| | TRI | -0.3296 | 0.0008 | 0.0081* |
| *Climatic* | ATR | -0.1864 | 0.0633 | 0.6335 |
| | PWQ | -0.5081 | 0.0000 | 0.0000* |
| | MAP | -0.5052 | 0.0000 | 0.0000* |
| | MAT | -0.2430 | 0.0149 | 0.1486 |

Abbreviations defined as follows: topographic wetness index (TWI), terrain ruggedness index (TRI), annual temperature range (ATR), precipitation of the warmest quarter (PWQ), mean annual precipitation (MAP), and mean annual temperature (MAT).

*Statistically significant. Adjusted p values are Bonferroni-corrected.

## Zhang et al.'s pH prediction CART model

In 2019, Zhang et al. developed a pH prediction CART model with topographic and climatic data collected from a hilly region of southwestern China [15]. The current study adapted Zhang et al.'s model to data collected within New York's Temperate Deciduous Forests to assess the geographic specificity of its pH predictions.

Zhang et al.'s model uniformly predicted New York Temperate Deciduous Forest soil pH to be 5.43. For all sampling locations, factor values determinative of predicted pH in Zhang et al.'s model measured consistently below or above its split cutoff values, despite variation within factor data. ATR exceeded 28.85 degrees Celsius, elevation fell below 1297 meters, and channel network values never equaled or surpassed 867.38 for all locations. Nevertheless, Zhang et al.'s CART was 'good' at predicting Temperature Deciduous Forest soil pH (% RRMSE = 14.53), although its predictions differed significantly from observed pH (p<0.001) and it experienced an increase in error relative to its estimations in southwestern China (% RRMSE = 14.53 as compared to % RRMSE = 6.9).

## Novel pH prediction models and factor importances

For alternative Temperate Deciduous Forest pH estimations, novel CART and RF models were generated. The novel CART model (Fig 2) 'excellently' predicted soil pH (% RRMSE = 9.33) and predictions did not differ significantly from actual data (p = 0.77).

The novel RF model also 'excellently' predicted Temperate Deciduous Forest soil pH (Fig 3). The RF model's pH estimations yielded a RRMSE 5% lower than that of the CART model (% RRMSE = 8.88) and its predictions did not differ significantly from observed pH (p = 0.07) or CART predictions (p = 0.26).

Plotting the mean squared error (MSE) of the RF model as individual decision trees were recursively added to the 'forest' demonstrates that n = 1000 trees was sufficient to minimize model MSE (Fig 4).

For both novel models, climatic factors were more important to model construction than topographic factors. The two most important factors for CART and RF model construction were mean annual temperature(CVI = 7.5, % IncMSE = 0.092) and mean annual precipitation (CVI = 7.21, % IncMSE = 0.052). The fourth most important factor for each model was also a

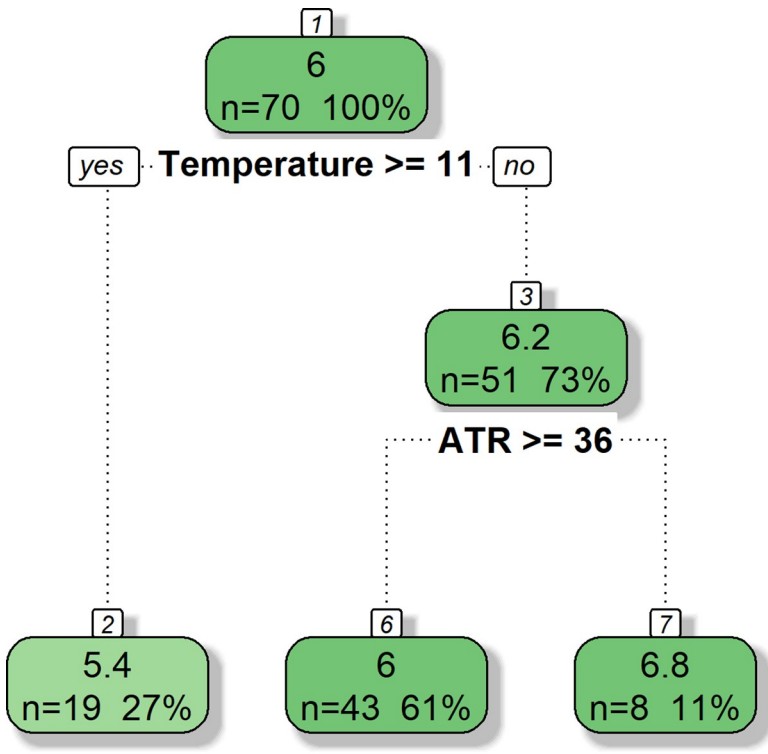

**Fig 2. Novel CART model predicting Temperate Deciduous Forest soil pH.** See Table 1 for descriptions of all variables.

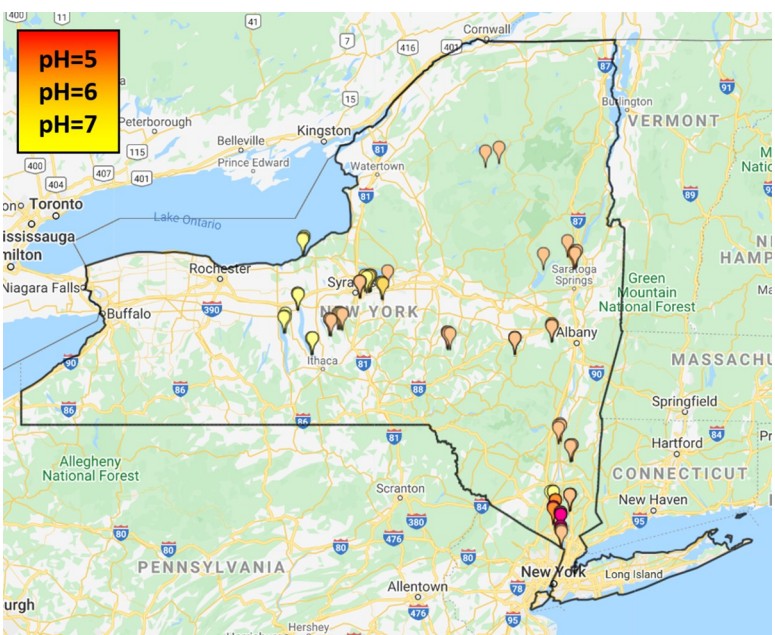

**Fig 3. Predicted pH by the random forest for all sampling locations.** Republished from [31] under a CC BY license, with permission from ZeeMaps, original copyright 2005.

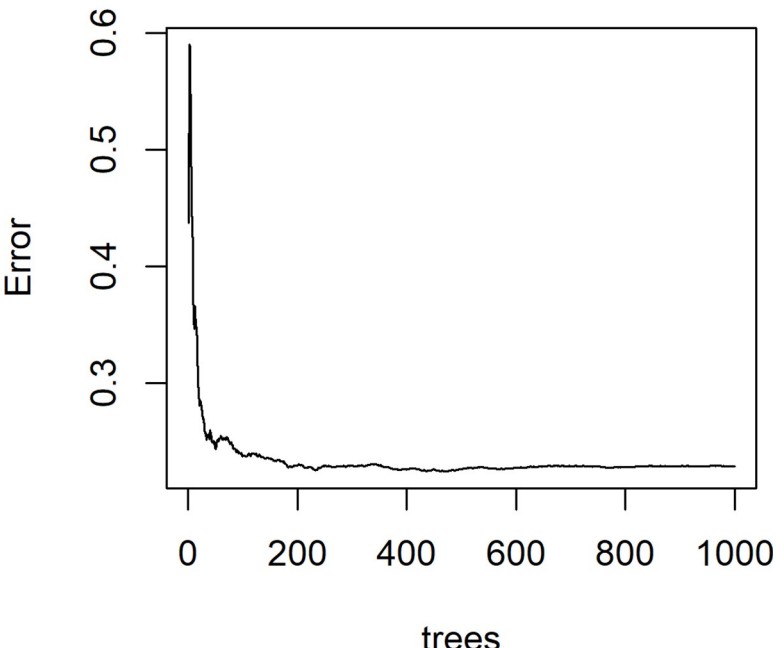

**Fig 4. Random forest model MSE by number of trees in the model (31).**

climatic factor, precipitation of the warmest quarter (CVI = 4.8, % IncMSE = 0.041). Additional factor importances are available in Table 3.

## Discussion

The current study investigated the extrapolative capacity of a previously developed CART model as compared to novel CART and random forest models for soil pH prediction. We

**Table 3. Factor importances to CART and RF model construction.**

| Factors | CVI | % IncMSE |
|---|:---:|:---:|
| MAT | 7.5[1] | 0.092[1] |
| MAP | 7.21[2] | 0.052[2] |
| PWQ | 4.80[4] | 0.041[4] |
| ATR | 5.31[3] | 0.029[6] |
| Elevation | 3.95[5/6] | 0.045[3] |
| Channel network | 3.95[5/6] | 0.024[5] |
| TRI | 0.42[7] | 0.004[9] |
| TWI | N/A | 0.008[7] |
| Slope | N/A | 0.005[8] |
| Valley depth | N/A | -0.001[10] |

Abbreviations defined as follows: topographic wetness index (TWI), terrain ruggedness index (TRI), annual temperature range (ATR), precipitation of the warmest quarter (PWQ), mean annual precipitation (MAP), and mean annual temperature (MAT). Reference S1 File for a definition of each factor.

[1,2,3,4,5,6,7,8,9,10]Rankings of variable importance from most important (1) to least important (10).

found that a model developed with data from southwestern China had higher predictive error when applied in our study region, supporting our first hypothesis regarding geographic specificity of CART models. A random forest model was not significantly more accurate at predicting soil pH than a CART model, disagreeing with our second hypothesis regarding the superiority of random forest models.

## Geographic specificity

Previously, Zhang et al. [15] developed a classification and regression tree for pH prediction with data from southwestern China. In the current study, the geographic specificity of Zhang et al.'s model was assessed via its application to data from New York's Temperate Deciduous Forests. Relying on this data, the model suffered an increase in predictive error from 6.9% RRMSE (in southwestern China) to 14.53% RRMSE (in New York), demonstrating its limited pH prediction ability in an alternative geographic region. When a novel CART model was constructed using Temperate Deciduous Forest data, it predicted pH more accurately than Zhang et al.'s model (% RRMSE = 9.33), demonstrating that CART pH predictions are most accurate in the geographic area from which their training data is sourced. As expected, the novel CART model predicted Temperature Deciduous Forest soil pH with an accuracy approximately equal to that of Zhang et al.'s CART model in southwestern China. The novel CART % RRMSE in New York, 9.3, is nearly equal to Zhang et al.'s CART RRMSE in China, 6.9.

Our approach of testing geography was novel and necessary to explore the limitations of soil pH model extrapolation. The failure of Zhang et al.'s model to transfer to a novel geographic region may be due to spatial autocorrelation between factors affecting soil pH. Existing literature documents that attributes close to one another in geographic space and time are similar to one another in value [53]. As a result, the distribution of ecological data in a region is different from the distribution in another and predictive models trained in one learn to fit its distribution specifically [54–56]. Research seeking to improve the extrapolative and interpolative abilities of ecological models has turned to accounting for spatial autocorrelation for this reason; specifically, in the fields of biodiversity conservation [57] and pedometrics [58–61]. Future research can apply these concepts to soil pH prediction by further testing the limitations of current models and identifying general rules that increase the likelihood of successful model application to new regions.

## CART vs. random forest models

We report that random forest models are not significantly more accurate at predicting soil pH than CART models. This disagrees with previous research identifying weaknesses of CART modelling relative to random forests [21, 55, 56, 62]. CART models are susceptible to overfitting training data and fail to maximize predictive accuracy because of unavoidable skewness in training data. Random forests, on the other hand, not only incorporate randomization and bootstrapping into the construction of individual Categorization and Regression trees, but also aggregate the output of many individual trees. These characteristics are thought to counter overfitting and generally improve accuracy [22, 24, 29, 63–67]. The current study disagrees with these findings in the context of soil pH prediction. We report that random forest models are not meaningfully more accurate at soil pH prediction from topographic and climatic factors than individual classification and regression trees for our study region. The RF % RRMSE, 8.88, is approximately equal to the CART RRMSE, 9.33, and model predictions did not differ significantly from one another (p = 0.26).

## Factors influencing soil pH

The second most important factor for both CART and RF model construction was the climatic factor mean annual precipitation (MAP). MAP demonstrated relatively high and statistically significant correlations with soil pH ($\rho$ = -0.51, p<0.001). Precipitation of the warmest quarter (PWQ) was the fourth most important factor to CART and RF model construction and also exhibited a high and significant correlation with pH ($\rho$ = -0.51, p<0.001).

These findings contrast with those of Zhang et al., who found that the annual temperature range (ATR), terrain wetness index (TWI) and Melton ruggedness number were most important to pH. These factors may have been more important in Zhang et al's study due to their analysis of a hilly region with much greater variability in elevation and slope than in the current study. The standard deviation of elevation and slope observed here were 140m and 5.9˚, less than the 254m and 8˚ observed by Zhang et al.

This interpretation is supported by previous literature. While global soil pH is highly influenced by precipitation [68], regional heterogeneities can enhance the influence of other factors. In some regions, south-facing slopes tend to exhibit more basic soil pH than north-facing slopes [69] and in others slope direction alone has no significant relationship to pH [70]. The strength of the association between elevation and pH also varies by region, ranging from r = -0.3 in a broadleaf forest [19] to r = -0.5 in a subtropical rainforest [17]. To this effect, Zhang et al. reported a correlation between soil pH and elevation with r = -0.014 [15]. It is probable that the unique topographical characteristics of Zhang et al.'s studied region, including its varied elevation and slope, are responsible for their CART model's emphasis on ATR, TWI and Melton ruggedness.

In future model creation, swapping factors that exhibited weak or nonexistent correlations to soil pH and that only minimally contributed to model construction with alternative topographic or climatic factors could improve predictive accuracy [24]. Previous literature has utilized linear regression [71] or Boruta all-relevant variable selection for this purpose [54, 72]. Model construction from the minimal-optimal set of variables reduces overfitting and increases interpretability [54].

## Applications

Provisioning and regulating services are the most important ecosystem services for security, the basic materials for a good life, and health (1). These services are provided for by supporting services, such as plant growth and diversity, that are controlled by soil pH (2). In this way, the ability to accurately predict global soil pH would expand the scientific knowledge of natural environmental regulation and facilitate an increase in the efficiency of agricultural production. These impacts could improve global living standards by countering climate change and mitigating world hunger.

Regarding climate change, soil carbon-sequestration by plant roots has been proposed as a mechanism for extracting $CO_2$ from the atmosphere [73]. However, the efficiency of root-mediated carbon storage varies by plant species [74] whose optimal growth is influenced by soil pH [75]. In this context, soil pH prediction models could be leveraged to identify ideal growth regions for the large-scale breeding of carbon-sequestering plant species.

Regarding world hunger, carbon-sequestration in soil is also expected to improve crop yields by replenishing historically depleted organic matter that is needed for plant growth [76]. Therefore, pH prediction models could inform agricultural workers about their ability to seed carbon-sequestering plant species on their land, replenish their soil's carbon content, and improve their crop yields. Further, rice growth is highly pH-dependent [77] and many developing regions rely on rice to provide a large portion of their populations' average caloric intake

[78]. Soil pH prediction models could be used to discern areas naturally conducive to rice growth and improve production. These concepts apply equally to other nourishing plant species.

As shown here, the accuracy of pH prediction, and the ability to leverage pH for combating climate change and world hunger, is influenced by the geographic specificity of predictive models. Therefore, to fully realize the potential of natural soils, our study encourages researchers to restrict the extrapolation of predictive models to the regions from which their training data was sourced.

Taken together, future research should seek to identify a combination of predictors that explain a larger fraction of pH variance, to construct alternative CART and RF pH prediction models for additional geographic regions, and to leverage these improved models for the seeding of carbon-sequestering and nutrient-providing plant species.

## Conclusions

Previous literature has established that CART modelling can be applied to soil pH prediction. The current study sought to address the extrapolative capacity of Zhang et al.'s pH prediction CART model in the Temperate Deciduous Forest relative to novel methods of pH prediction in this region. Results indicated that Zhang et al.'s CART model experienced a reduction in predictive accuracy when applied to data from the Temperate Deciduous Forest and was outperformed by novel CART and RF models. We report that pH prediction models are most accurate when applied to their training data's geographic region, that RF modeling provides no notable advantage over CART modeling in the realm of soil pH prediction, and that climatic factors are useful for model construction.

## Supporting information

**S1 File. Topographic and climatic factor information.**
(XLSX)

**S2 File. Topographic and climatic factor and soil pH data.**
(XLSX)

## Acknowledgments

The authors appreciate the help of Mr. Jeffrey Wuebber who has provided guidance for us throughout our early scientific careers. Thank you to Mr. Egelberg and Mr. and Mrs. Pena for assisting with travel and pH data collection.

## Author Contributions

**Conceptualization:** Jacob Egelberg, Christina Andruk.

**Data curation:** Jacob Egelberg.

**Formal analysis:** Jacob Egelberg.

**Investigation:** Jacob Egelberg.

**Methodology:** Jacob Egelberg, Nina Pena.

**Project administration:** Jacob Egelberg.

**Resources:** Jacob Egelberg.

**Supervision:** Jacob Egelberg, Christina Andruk.

**Validation:** Rachel Rivera.

**Writing – original draft:** Jacob Egelberg, Rachel Rivera.

**Writing – review & editing:** Jacob Egelberg, Rachel Rivera.

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
