## [Decision Letter · Decision Letter 0]

27 Apr 2021

PONE-D-21-10198

Assessing the geographic specificity of pH prediction by classification and regression trees

PLOS ONE

Dear Dr. Egelberg,

Thank you for submitting your manuscript to PLOS ONE. After careful consideration, we feel that it has merit but does not fully meet PLOS ONE’s publication criteria as it currently stands. Therefore, we invite you to submit a revised version of the manuscript that addresses the points raised during the review process.

The submitted manuscript has some merit but I totally agree with both reviewers that this work needs some major revisions before being considered to be published in PLON ONE. I'm mainly concern about the short description of the state of the art (Introduction) and the methods section. refer 2 highlighted this point in the revisions,

Other aspect is that I've some difficulty  to the highly the scientific novelty of this work and the future applications. The authors should clearly highlight this aspect.

We look forward to receiving your revised manuscript.

Kind regards,

João Canário, PhD

Academic Editor

PLOS ONE

Journal Requirements:

2. We note that Figures 1 and 3 in your submission contain map images which may be copyrighted.

a. You may seek permission from the original copyright holder of Figures 1 and 3 to publish the content specifically under the CC BY 4.0 license. 

Reviewers' comments:

Reviewer's Responses to Questions

**Comments to the Author**

1. Is the manuscript technically sound, and do the data support the conclusions?

Reviewer #1: Yes

Reviewer #2: Partly

2. Has the statistical analysis been performed appropriately and rigorously? 

Reviewer #1: Yes

Reviewer #2: Yes

3. Have the authors made all data underlying the findings in their manuscript fully available?

Reviewer #1: Yes

Reviewer #2: Yes

4. Is the manuscript presented in an intelligible fashion and written in standard English?

Reviewer #1: Yes

Reviewer #2: Yes

5. Review Comments to the Author

Reviewer #1: The manuscript entitled “Assessing the geographic specificity of pH prediction by classification and regression trees” by Egelberg et al. showed the influence of different geographic conditions in the prediction of pH by CART´s (classification and regression trees), their limitations by machine learning and the use of new CART´s and random forest (RF) models to compare with previous ones applied in different regions. This study is based on a previous one conducted by Zhang et al. (2019) and the authors applied the CART´s model developed by Zhang et al. and compared with new ones specially used for this study.

General comments:

The manuscript is well written, the introduction is short but contain all the necessary information for the current study. The goals of the study are simple and well designated, very understandable and in agreement with what the authors want to understand beyond what is already known. Material and methods section is well designed with necessary detail. The results are well presented and divided into appropriate key points. The discussion section is in accordance with the presented goals and results from the study and the conclusions are well described, according to the discussion of the obtained results and aligned with the hypotheses presented in the goals and tested with the results.

With a short revision (indicated below in the specific comments and considered as minor revision) in order to improve the manuscript, I recommend this research for publication.

Specific comments:

Material and Methods:

Line 103: Caption of Fig. 1: where is “Created” should be “created”.

Line 107: dH20 is deionized water? If yes, please indicate in the text; “…vigorously shaken…” during which time? And how? Please indicate!

Line 143: Please describe each of the parameters in the equation (1).

Line 148: Where is (2), (44) indicate as (2, 44). Are both references right?

Results:

Lines 167-173: you should indicate the meaning of all abbreviations used in the table and not only of those discussed. You can add this information in the end of the table as notes.

Line 177: According to the table 3 the percentage of 26.6 is actually 26.4 (see table 3). please correct accordingly in the text.

Line 205: Since the manuscript only have 3 figures I think that you can use this figure in S2 in the manuscript too, as fig. 4.

Reviewer #2: Comments on the manuscript PONE-D-21-10198

The manuscript “Assessing the geographic specificity of pH prediction by classification and regression trees” by Egelberg et al. intends to evaluate the suitability of machine learning on the prediction of pH soil, by replicating the approach of Zhang et al. (2019) using CART model and by complementing it with random forest models.

The manuscript has a good structure and it is easy to read. However, I do find some weaknesses that should be revised to improve the paper. My major concern is related with the overall lack of detail in the manuscript. The hypothesis should be clearly indicated. The discussion in particular lacks an integration of data obtained, whereas the authors provide general information without discussing their data. Other comments are also indicated.

I suggest the authors a major revision of the manuscript.

Aims

Lines 91-93 should precede the aims. Can the authors give more detail on the accuracy of RF prediction?

Study area

Is there any information that could be added to describe pH in the soils sampled?

Soil pH testing

Can the authors provide a reference for the procedure adopted?

The use of pH strips does not provide sufficient accuracy for the measurements, which could benefit the model approach used by the authors.

Topographic and climatic data

Table 1 is merely informative and in part replicates tables 2 and 3. It could be in supporting information since it does not provide substantial information relevant to study.

Statistical analysis

Did the authors check the normality of the data?

Lines 167-168: to assess ‘the strength of the linear association between variables and soil pH’ using Pearson correlation the authors have to indicate if variables did present normal distribution.

Discussion

Lines 219-221: This is a conclusion drawn from your work. Please, revise the start of your discussion.

The discussion lacks the support of the data obtained.

6. PLOS authors have the option to publish the peer review history of their article (what does this mean?). If published, this will include your full peer review and any attached files.

Reviewer #1: No

Reviewer #2: No

---

## [Author Response · Author response to Decision Letter 0]

26 May 2021

RESPONSE TO EDITOR:

EDITOR COMMENT: The submitted manuscript has some merit but I totally agree with both reviewers that this work needs some major revisions before being considered to be published in PLON ONE. I'm mainly concern about the short description of the state of the art (Introduction) and the methods section. refer 2 highlighted this point in the revisions, Other aspect is that I've some difficulty to the highly the scientific novelty of this work and the future applications. The authors should clearly highlight this aspect.

RESPONSE: Thank you for coordinating the review process. We have revised the

manuscript and addressed Reviewer’s comments. To address your recommendations, we have elaborated on the importance of our research in the Discussion section.

General Comments:

COMMENT: Please ensure that your manuscript meets PLOS ONE's style requirements, including those for file naming.

RESPONSE: We have checked that all formatting and naming requirements have been met.

COMMENT: We note that Figures 1 and 3 in your submission contain map images which may be copyrighted. We require you to either (a) present written permission from the copyright holder to publish these figures specifically under the CC BY 4.0 license, or (b) remove the figures from your submission:

RESPONSE: We have obtained permission from the copyright holder, ZeeMaps, to publish their images. A PLOS ONE Content Permission Form has been uploaded.

 

RESPONSE TO REVIEWER 1:

REVIEWER COMMENT: The manuscript is well written, the introduction is short but contain all the necessary information for the current study. The goals of the study are simple and well designated, very understandable and in agreement with what the authors want to understand beyond what is already known. Material and methods section is well designed with necessary detail. The results are well presented and divided into appropriate key points. The discussion section is in accordance with the presented goals and results from the study and the conclusions are well described, according to the discussion of the obtained results and aligned with the hypotheses presented in the goals and tested with the results. With a short revision (indicated below in the specific comments and considered as minor revision) in order to improve the manuscript, I recommend this research for publication.

RESPONSE: We thank the reviewer for their kind words. We have revised our manuscript according to their recommendations below.

REVIEWER COMMENT: Line 103: Caption of Fig. 1: where is “Created” should be “created”.

RESPONSE: We have revised the caption of Fig. 1 to read “Study area. Points indicate state parks where samples were collected. Created with ZeeMaps.”

REVIEWER COMMENT: Line 107: dH20 is deionized water? If yes, please indicate in the text; “…vigorously shaken…” during which time? And how? Please indicate!

RESPONSE: We specified this line to read “deionized H2O.” We replaced the phrasing of “vigorously shaken” with a more specific description: “The mixture was vigorously inverted and rotated before being let to sit for 10 minutes…” 

REVIEWER COMMENT: Please describe each of the parameters in the equation (1).

RESPONSE: We added definitions of y, y ^, and y ®, as well as an in-text description of the equation.

REVIEWER COMMENT: Line 148: Where is (2), (44) indicate as (2, 44). Are both references right?

RESPONSE: We redid notation to clearly distinguish between references to a citation and references to an equation. References to an equation are now labeled and in brackets.

REVIEWER COMMENT: Lines 167-173: you should indicate the meaning of all abbreviations used in the table and not only of those discussed. You can add this information in the end of the table as notes.

RESPONSE: We added all abbreviations in the subsection Topographic and climatic data.

REVIEWER COMMENT: Line 177: According to the table 3 the percentage of 26.6 is actually 26.4 (see table 3). please correct accordingly in the text.

RESPONSE: We re-constructed Table 3 with data from Spearman correlations rather than Pearson correlations. We performed this step to accommodate reviewer 2’s methodological recommendations. Updated values in the table match those in the text, but the specific sentence identified here was removed because it is not appropriate to square the Spearman Rho. 

REVIEWER COMMENT: Line 205: Since the manuscript only have 3 figures I think that you can use this figure in S2 in the manuscript too, as fig. 4.

RESPONSE: We have added figure S2 in to the manuscript as figure 4.

RESPONSE TO REVIEWER 2:

REVIEWER COMMENT: The manuscript “Assessing the geographic specificity of pH prediction by classification and regression trees” by Egelberg et al. intends to evaluate the suitability of machine learning on the prediction of pH soil, by replicating the approach of Zhang et al. (2019) using CART model and by complementing it with random forest models.

The manuscript has a good structure and it is easy to read. However, I do find some weaknesses that should be revised to improve the paper. My major concern is related with the overall lack of detail in the manuscript. The hypothesis should be clearly indicated. 

RESPONSE: See subsection Aim of the current study in the Introduction for our hypotheses. 

The discussion in particular lacks an integration of data obtained, whereas the authors provide general information without discussing their data. 

RESPONSE: Data from the Results was incorporated into the Discussion to support each claim made. 

Other comments are also indicated. I suggest the authors a major revision of the manuscript.

RESPONSE: We thank the reviewer for their effort and attention to detail. We have revised our manuscript according to their recommendations below.

REVIEWER COMMENT: Lines 91-93 should precede the aims. Can the authors give more detail on the accuracy of RF prediction?

RESPONSE: We believe the logical flow of the paper is better maintained by stating the study objectives prior to our expectations regarding those objectives. As such, we have kept lines 91-93 before the aims. This decision aligns with the recommendation of Schober et al. in “Clear Study Aims and Hypotheses in a Research Paper.” 

However, we updated our hypotheses to provide a more specific estimate regarding the accuracy of RF prediction. Because we believed that the RF would be more accurate than the CART, and that our CART would demonstrate an accuracy in our region similar to Zhang et al.’s 6.9% RRMSE in their region, we hypothesized that our RF would yield a % RRMSE of below 6.9. 

REVIEWER COMMENT: Is there any information that could be added to describe pH in the soils sampled?

RESPONSE: We added a statement describing the observed pH range in our study region. 

REVIEWER COMMENT: Can the authors provide a reference for the procedure adopted?

RESPONSE: We added a reference from which we derived our protocol. 

REVIEWER COMMENT: The use of pH strips does not provide sufficient accuracy for the measurements, which could benefit the model approach used by the authors.

RESPONSE: We updated the description of our methodology with additional details and citations to support its robustness. Specifically, we added that we used four-squared pH test strips for pH testing. Previous literature has found these types of test strips to exhibit greater than 95% positive and negative predictive power and greater than 90% sensitivity and specificity in moderately acidic pH. Considering that our observed pH range falls within this moderately acidic category, four-squared plastic test strips were appropriate in the current context. 

REVIEWER COMMENT: Table 1 is merely informative and in part replicates tables 2 and 3. It could be in supporting information since it does not provide substantial information relevant to study.

RESPONSE: We moved Table 1 to supporting information.

REVIEWER COMMENT: Did the authors check the normality of the data?

RESPONSE: We did not check for normality; however, all sample sizes exceed 30 samples. By the central limit theorem, this implies that the sampling distribution of our sample means for all factors is approximately normal and t-tests are appropriate. We applied Levene’s test for homogeneity of variance found variances to differ between samples. As such, we applied a two-tailed Welch’s t-test to our data. We added citations to our manuscript to support our analysis.

REVIEWER COMMENT: Lines 167-168: to assess ‘the strength of the linear association between variables and soil pH’ using Pearson correlation the authors have to indicate if variables did present normal distribution.

RESPONSE: We recalculated Pearson correlations with the Spearman correlation, a rank-based method that does not assume normality. This recalculation altered the strength and significance of the relationship between some factors and soil pH, but our initial conclusions remain largely intact. Changes have been clearly noted in the revised manuscript. We thank the reviewer for bringing this assumption of the Pearson correlation to our attention and have added a citation that discusses the appropriate use of correlation coefficients to support our analysis and as a resource for future readers. 

Further, though this was not requested by the reviewer, in our recalculation of correlational significance we adjusted our p-values with the Bonferroni correction. Because we performed multiple statistical tests on different characteristics of the same samples, the global probability of a type I error is inflated. To counter this, we multiplied p-values by the number of tests performed as suggested by Bonferroni. This is mathematically equivalent to dividing the probability of a type I error (α) by the number of tests. We added references to support this decision. We did not perform this step in the original manuscript because the authors were made aware of the Bonferroni correction after our initial submission.

REVIEWER COMMENT: Lines 219-221: This is a conclusion drawn from your work. Please, revise the start of your discussion.

RESPONSE: We agree that this is a conclusion drawn from our work, but we dispute that this is problematic to include in this section. We believe it is important to address the accuracy of our hypotheses upfront in the Discussion and we believe that our conclusions are relevant to our discussion of the results. 

To clarify the flow of our Discussion, we have split it into subsections. Each of these subsections interprets a portion of our results in the context of our initial hypotheses. We have also expanded the interpretation of our results in the context of previous literature.

---

## [Decision Letter · Decision Letter 1]

22 Jun 2021

PONE-D-21-10198R1

Assessing the geographic specificity of pH prediction by classification and regression trees

PLOS ONE

Dear Dr. Egelberg,

Thank you for submitting your manuscript to PLOS ONE. After careful consideration, we feel that it has merit but does not fully meet PLOS ONE’s publication criteria as it currently stands. Therefore, we invite you to submit a revised version of the manuscript that addresses the points raised during the review process.

After a careful reading of the reviewers comments (mainly reviewer 2) i tend to agree that the manuscript needs some clarification before being accepted for publication in PLOS ONE. While reviewer 2 comment 1 and 4 are a question of style I tend to corroborate both comments, suggesting a revision accordingly.

Comments 2 and 3 from the same reviewer are more delicate and needs a clear clarification. I agree with the reviewer that the normality of the data should be checked. Also the Pearson and Spearman correlations issue should be clarified.

We look forward to receiving your revised manuscript.

Kind regards,

João Canário, PhD

Academic Editor

PLOS ONE

Journal Requirements:

Reviewers' comments:

Reviewer's Responses to Questions

**Comments to the Author**

1. If the authors have adequately addressed your comments raised in a previous round of review and you feel that this manuscript is now acceptable for publication, you may indicate that here to bypass the “Comments to the Author” section, enter your conflict of interest statement in the “Confidential to Editor” section, and submit your "Accept" recommendation.

Reviewer #1: All comments have been addressed

Reviewer #2: (No Response)

2. Is the manuscript technically sound, and do the data support the conclusions?

Reviewer #1: Yes

Reviewer #2: Partly

3. Has the statistical analysis been performed appropriately and rigorously? 

Reviewer #1: Yes

Reviewer #2: No

4. Have the authors made all data underlying the findings in their manuscript fully available?

Reviewer #1: Yes

Reviewer #2: Yes

5. Is the manuscript presented in an intelligible fashion and written in standard English?

Reviewer #1: Yes

Reviewer #2: Yes

6. Review Comments to the Author

Reviewer #1: taking into account that the authors made all the changes proposed and that the manuscript was improved with the help of the comments of both reviewers, I considered that this paper can be published in the present form.

Reviewer #2: REVIEWER 2 COMMENTS TO THE AUTHOR’S RESPONSE:

The manuscript “Assessing the geographic specificity of pH prediction by classification and regression trees” by Egelberg et al. was revised according to the previous reviewer 2’ comments. However, some details need to be clarified. Consequently, I suggest the author to revise the manuscript in some critical issues, in particular points 1, 2 and 4, as follows:

1- AUTHOR’S RESPONSE: We believe the logical flow of the paper is better maintained by stating the study objectives prior to our expectations regarding those objectives. As such, we have kept lines 91-93 before the aims. This decision aligns with the recommendation of Schober et al. in “Clear Study Aims and Hypotheses in a Research Paper.”

Reviewer comment: I strongly disagree with the logical argument provided by the authors. Contrarily to what is stated by the authors, one can only postulate objectives after elaborating a hypothesis, which is completely different from a hypothesis elaborated after the definition of objectives. Therefore, lines 93-97 of the revised manuscript should precede the definition of the aims of the study. Ultimately, the response given by the authors corroborates my observation.

2- AUTHOR’S RESPONSE: We did not check for normality; however, all sample sizes exceed 30 samples. By the central limit theorem, this implies that the sampling distribution of our sample means for all factors is approximately normal and t-tests are appropriate. We applied Levene’s test for homogeneity of variance found variances to differ between samples. As such, we applied a two-tailed Welch’s t-test to our data. We added citations to our manuscript to support our analysis.

Reviewer comment: The CLT has some disadvantages and is a common misconception if we take a large number of samples their distribution will be (close to) normal. Not everything is a mean. Therefore, it should be used very carefully when dealing with environmental samples. As an example, if you consider two vertical profiles, e.g. for pH, between 1 to 10-cm depth and one presents the highest value at the surface (1-cm) and other presents the same highest value at 6-cm, their distributions are completely different despite having the same average value. This is even more complicated when such variable depends on other parameters of the soils, as it is the case of this work when looking into a spatial distribution. Thus, normality should be checked using the appropriate statistical test and indicated in the Methods for the sake of the reader withdraw its own conclusions.

3- AUTHOR’S RESPONSE: We recalculated Pearson correlations with the Spearman correlation, a rank-based method that does not assume normality.

Reviewer comment: Please be aware that Pearson and Spearman correlations have different assumptions, so I assume the authors meant to say they have recalculated correlations.

4- AUTHOR’S RESPONSE: We agree that this is a conclusion drawn from our work, but we dispute that this is problematic to include in this section. We believe it is important to address the accuracy of our hypotheses upfront in the Discussion and we believe that our conclusions are relevant to our discussion of the results.

Reviewer comment: I do not understand or agree with this logic, as was earlier expresses for the objectives and hypothesis. One can discuss their findings and subsequently draw the conclusions but to discuss results based in conclusions is not logical and scientifically incorrect.

7. PLOS authors have the option to publish the peer review history of their article (what does this mean?). If published, this will include your full peer review and any attached files.

Reviewer #1: No

Reviewer #2: No

---

## [Author Response · Author response to Decision Letter 1]

6 Jul 2021

RESPONSE TO EDITOR:

EDITOR COMMENT: After careful consideration, we feel that it has merit but does not fully meet PLOS ONE’s publication criteria as it currently stands. Therefore, we invite you to submit a revised version of the manuscript that addresses the points raised during the review process. After a careful reading of the reviewers comments (mainly reviewer 2) i tend to agree that the manuscript needs some clarification before being accepted for publication in PLOS ONE. While reviewer 2 comment 1 and 4 are a question of style I tend to corroborate both comments, suggesting a revision accordingly. Comments 2 and 3 from the same reviewer are more delicate and needs a clear clarification. I agree with the reviewer that the normality of the data should be checked. Also the Pearson and Spearman correlations issue should be clarified.

RESPONSE: Thank you for your comments. We have addressed your recommendations with our responses to Reviewer 2’s input. 

GENERAL COMMENTS:

COMMENT: Please review your reference list to ensure that it is complete and correct. If you have cited papers that have been retracted, please include the rationale for doing so in the manuscript text, or remove these references and replace them with relevant current references. Any changes to the reference list should be mentioned in the rebuttal letter that accompanies your revised manuscript. If you need to cite a retracted article, indicate the article’s retracted status in the References list and also include a citation and full reference for the retraction notice.

RESPONSE: We have reviewed our references and can ensure that it is complete and correct. There are no retracted papers included in our references.

 

RESPONSE TO REVIEWER 1:

REVIEWER COMMENT: Taking into account that the authors made all the changes proposed and that the manuscript was improved with the help of the comments of both reviewers, I considered that this paper can be published in the present form.

RESPONSE: We thank the reviewer for their time and consideration. 

RESPONSE TO REVIEWER 2:

REVIEWER COMMENT: The manuscript “Assessing the geographic specificity of pH prediction by classification and regression trees” by Egelberg et al. was revised according to the previous reviewer 2’ comments. However, some details need to be clarified. Consequently, I suggest the author to revise the manuscript in some critical issues, in particular points 1, 2 and 4, as follows.

RESPONSE: We thank the reviewer for their additional input and address each recommendation below.

REVIEWER COMMENT: I strongly disagree with the logical argument provided by the authors. Contrarily to what is stated by the authors, one can only postulate objectives after elaborating a hypothesis, which is completely different from a hypothesis elaborated after the definition of objectives. Therefore, lines 93-97 of the revised manuscript should precede the definition of the aims of the study. Ultimately, the response given by the authors corroborates my observation.

RESPONSE: We reformatted our objectives and hypothesis so that statement of the objectives follows statement of the hypotheses. We trust the reviewer’s judgement that this improves clarity.

REVIEWER COMMENT: The CLT has some disadvantages and is a common misconception if we take a large number of samples their distribution will be (close to) normal. Not everything is a mean. Therefore, it should be used very carefully when dealing with environmental samples. As an example, if you consider two vertical profiles, e.g. for pH, between 1 to 10-cm depth and one presents the highest value at the surface (1-cm) and other presents the same highest value at 6-cm, their distributions are completely different despite having the same average value. This is even more complicated when such variable depends on other parameters of the soils, as it is the case of this work when looking into a spatial distribution. Thus, normality should be checked using the appropriate statistical test and indicated in the Methods for the sake of the reader withdraw its own conclusions.

RESPONSE: To test for normality, we plotted histograms of predicted and actual pH data with the hist() function in R. CART and random forest-predicted pH values appear normally distributed; however, sampled pH and Zhang et al.’s predicted pH values do not. 

To air on the side of caution and avoid an improper analysis, we recalculated significance with non-parametric Wilcoxon Rank Sum tests that do not assume a normal sampling distribution of sample means. The recalculation of significance did not alter our results or conclusions. 

REVIEWER COMMENT: Please be aware that Pearson and Spearman correlations have different assumptions, so I assume the authors meant to say they have recalculated correlations.

RESPONSE: We agree that the Pearson and Spearman correlations measure fundamentally different phenomena and intended to state that we replaced calculations of the Pearson correlation with calculations of the Spearman correlation in our analysis. We altered language in our paper to account for these divergent assumptions. Specifically, we removed R^(2) values, replaced r values produced by the Pearson correlation with rho values produced by the Spearman correlation, and reworded our statistical analysis section to emphasize that Spearman correlations measure the strength of monotonic relationships (as opposed to Pearson correlations, which are only valid for linear relationships). 

REVIEWER COMMENT: I do not understand or agree with this logic, as was earlier expresses for the objectives and hypothesis. One can discuss their findings and subsequently draw the conclusions but to discuss results based in conclusions is not logical and scientifically incorrect.

RESPONSE: We have reworded the introduction to our discussion in accordance with the reviewers’ comments.

---

## [Decision Letter · Decision Letter 2]

12 Jul 2021

Assessing the geographic specificity of pH prediction by classification and regression trees

PONE-D-21-10198R2

Dear Dr. Egelberg,

We’re pleased to inform you that your manuscript has been judged scientifically suitable for publication and will be formally accepted for publication once it meets all outstanding technical requirements.

Kind regards,

João Canário, PhD

Academic Editor

PLOS ONE

Reviewers' comments:

Reviewer's Responses to Questions

**Comments to the Author**

1. If the authors have adequately addressed your comments raised in a previous round of review and you feel that this manuscript is now acceptable for publication, you may indicate that here to bypass the “Comments to the Author” section, enter your conflict of interest statement in the “Confidential to Editor” section, and submit your "Accept" recommendation.

Reviewer #2: All comments have been addressed

2. Is the manuscript technically sound, and do the data support the conclusions?

Reviewer #2: Yes

3. Has the statistical analysis been performed appropriately and rigorously? 

Reviewer #2: Yes

4. Have the authors made all data underlying the findings in their manuscript fully available?

Reviewer #2: Yes

5. Is the manuscript presented in an intelligible fashion and written in standard English?

Reviewer #2: Yes

6. Review Comments to the Author

Reviewer #2: (No Response)

7. PLOS authors have the option to publish the peer review history of their article (what does this mean?). If published, this will include your full peer review and any attached files.

Reviewer #2: No

---

## [Editor Report · Acceptance letter]

16 Jul 2021

PONE-D-21-10198R2 

Assessing the geographic specificity of pH prediction by classification and regression trees 

Dear Dr. Egelberg:

I'm pleased to inform you that your manuscript has been deemed suitable for publication in PLOS ONE. Congratulations! Your manuscript is now with our production department. 

Kind regards, 

on behalf of

Dr. João Canário 

Academic Editor

PLOS ONE